# Microglial FABP4-UCP2 Axis Modulates Neuroinflammation and Cognitive Decline in Obese Mice

**DOI:** 10.3390/ijms23084354

**Published:** 2022-04-14

**Authors:** Simon W. So, Kendra M. Fleming, Cayla M. Duffy, Joshua P. Nixon, David A. Bernlohr, Tammy A. Butterick

**Affiliations:** 1Minneapolis Veterans Affairs Health Care System, Minneapolis, MN 55417, USA; soxxx063@umn.edu (S.W.S.); flemi169@umn.edu (K.M.F.); duffy024@umn.edu (C.M.D.); nixon049@umn.edu (J.P.N.); 2Department of Neuroscience, University of Minnesota Twin Cities, Minneapolis, MN 55455, USA; 3Department of Food Science and Nutrition, University of Minnesota Twin Cities, St. Paul, MN 55108, USA; 4Department of Surgery, University of Minnesota Medical School, Minneapolis, MN 55455, USA; 5Department of Biochemistry, University of Minnesota Twin Cities, Minneapolis, MN 55455, USA; bernl001@umn.edu

**Keywords:** FABP4, UCP2, microglia, cognitive decline, neuroinflammation, obesity

## Abstract

The microglial fatty-acid-binding protein 4-uncoupling protein 2 (FABP4-UCP2) axis is a key regulator of neuroinflammation in high-fat-diet (HFD)-fed animals, indicating a role for FABP4 in brain immune response. We hypothesized that the FABP4-UCP2 axis is involved in regulating diet-induced cognitive decline. We tested cognitive function in mice lacking microglial FABP4 (AKO mice). Fifteen-week-old male AKO and wild-type (WT) mice were maintained on 60% HFD or normal chow (NC) for 12 weeks. Body composition was measured using EchoMRI. Locomotor activity, working memory, and spatial memory were assessed using behavioral tests (open field, T-maze, and Barnes maze, respectively). Hippocampal microgliosis was assessed via immunohistochemical staining. An inflammatory cytokine panel was assayed using hippocampal tissue. Real-time RT-PCR was performed to measure microglial UCP2 mRNA expression. Our data support that loss of FABP4 prevents cognitive decline in vivo. HFD-fed WT mice exhibited impaired long- and short-term memory, in contrast with HFD-fed AKO mice. HFD-fed WT mice had an increase in hippocampal inflammatory cytokine expression (IFNγ, IL-1β, IL-5, IL-6, KC/GRO(CXCL1), IL-10, and TNFα) and microgliosis, and decreased microglial UCP2 expression. HFD-fed AKO mice had decreased hippocampal inflammatory cytokine expression and microgliosis and increased microglial UCP2 expression compared to HFD-fed WT mice. Collectively, our work supports the idea that the FABP4-UCP2 axis represents a potential therapeutic target in preventing diet-induced cognitive decline.

## 1. Introduction

Obesity is often characterized by chronic low-level inflammation, paralleling the prevalence of metabolic disorders such as type-2 diabetes mellitus [1,2,3]. Chronic neuroinflammation is often characterized by the increase in inflammatory cytokine release by microglia (brain immune cells) [4,5]. Chronic neuroinflammation and subsequent neuronal loss may be a key factor in the linkage between obesity and cognitive decline [6,7,8,9,10]. Overconsumption of high-fat diets (HFD), specifically those rich in saturated fatty acid (SFA) palmitic acid (PA), exacerbate neuroinflammation, neurodegeneration, and cognitive impairment [1,3,6,7,9,10,11,12,13]. Rodent studies indicate that this is influenced more by dietary SFA content than by total calories consumed [1,7]. Dietary PA increases the activation of a neurotoxic phenotype in microglia, resulting in a release of neurotoxic cytokines that may contribute to cognitive decline [3,14,15]. For example, chronic activation of microglia to a pro-inflammatory state can lead to reduced hippocampal cognitive function and neurogenesis [16,17,18,19]. While neuroinflammation and cognitive impairment are correlated, the contribution and the role of microglia in the context of obesity is undefined.

The link between inflammation and lipid metabolism indicates a key role for fatty acid-binding protein 4 (FABP4; also known as adipocyte protein 2; aP2) in neuroinflammatory diseases such as obesity and Alzheimer’s disease (AD). Importantly, molecular, genetic, or pharmacologic loss of FABP4 results in an anti-inflammatory phenotype preventing the development of metabolic syndrome or neuroinflammatory response even in the presence of an HFD [20,21]. With the inhibition or absence of FABP4, an increase in the mitochondrial membrane protein uncoupling protein 2 (UCP2) attenuates the production of reactive oxygen species (ROS) and prevents the switch to a pro-inflammatory state in both peripheral macrophages and brain immune cells such as microglia [20,21,22]. We hypothesized that knockout of FABP4 attenuates HFD-induced cognitive decline via reduced neuroinflammation characterized by microgliosis and inflammatory cytokine expression. In vivo, we demonstrate that mice maintained on an HFD have increased body fat percentage and reduced locomotor activity compared to mice maintained on normal chow (NC). Additionally, wild-type (WT) mice maintained on HFD have impaired long-term and short-term memory, as well as increased hippocampal cytokine expression and microgliosis, accompanied by a decrease in microglial UCP2 mRNA expression. Further, these effects are negated in mice lacking FABP4 (AKO). AKO mice have been shown to lack FABP4 mRNA and protein expression [23]. We demonstrate that AKO mice maintained on HFD have improved long and short-term memory, decreased hippocampal cytokine expression, decreased hippocampal microgliosis, and increased microglial UCP2 mRNA expression compared to HFD-fed WT mice.

## 2. Results

### 2.1. HFD Increases Fat Mass, Lean Mass, Body Weight, Fat/Lean Ratio, and Body Fat Percentage

To determine if HFD-fed AKO mice had altered changes in body composition compared to HFD-fed WT mice, we measured the body composition of mice using EchoMRI. We demonstrate that HFD-fed mice had increased fat mass (*p* < 0.0001; Figure 1A), lean mass (*p* < 0.013; Figure 1B), body weight (*p* < 0.0001; Figure 1C), fat/lean ratio (*p* < 0.0001; Figure 1D), and body fat percentage (*p* < 0.0001; Figure 1E) compared to NC-fed mice. We also demonstrate that HFD-fed AKO mice had slight but significant decreased fat mass (*p* < 0.044; Figure 1A), body weight (*p* < 0.0001; Figure 1C), fat/lean ratio (*p* < 0.0001; Figure 1D), and body fat percentage (*p* < 0.029; Figure 1E) compared to HFD-fed WT mice.

### 2.2. HFD Reduces Locomotor Activity

Prior studies indicate that HFD reduces locomotor activity [24,25]. In order to determine the effects of HFD on locomotor activity, mice were placed in an activity chamber for 2 h to measure general movement. We demonstrate that mice maintained on an HFD move significantly less compared to those maintained on NC, independent of genotype (*p* < 0.01; Figure 2B).

### 2.3. Loss of FABP4 Protects against Diet-Induced Cognitive Decline

To determine the effects of FABP4 on cognition, WT and AKO mice maintained on HFD or NC underwent various cognitive tasks. Learning was apparent in all groups in the Barnes maze task (*p* < 0.0001; Figure 3B). AKO mice maintained on an HFD have attenuated HFD-induced memory impairment as measured by a reduced latency to identify the target hole during the probe test in the Barnes maze task (Figure 3C). We demonstrate that WT mice maintained on an HFD have impaired memory as measured by increased latency to identify the target hole during the Barnes maze probe test (Figure 3C). We also demonstrated that HFD-fed WT mice had reduced alternations in T-maze, and this response is attenuated in HFD-fed AKO mice (Figure 4B).

### 2.4. Loss of FABP4 Leads to Decrease in Hippocampal IBA1 Expression in Mice Fed HFD

To determine if HFD-fed AKO mice had altered microglial activation in the hippocampus, we performed immunohistochemical staining for ionized calcium-binding adaptor molecule 1 (IBA1) in hippocampal tissues. We demonstrate that HFD-fed WT mice have increased hippocampal IBA1 expression compared to NC-fed mice (*p* < 0.001; Figure 5). We then demonstrate that HFD-fed AKO mice have decreased hippocampal IBA1 expression compared to HFD-fed WT mice (*p* < 0.02; Figure 5).

### 2.5. Loss of FABP4 Protects against HFD-Induced Increase in Hippocampal Inflammatory Cytokine Expression

To determine if HFD-fed AKO mice had an altered inflammatory cytokine profile in the hippocampus, we performed a multiplex inflammatory marker screening in hippocampal tissues. We demonstrate that HFD-fed WT mice have increased IFNγ (*p* < 0.0001; Figure 6A), IL-1β (*p* < 0.0001; Figure 6B), IL-5 (*p* < 0.0001; Figure 6C), IL-6 (*p* < 0.008; Figure 6D), KC/GRO(CXCL1) (*p* < 0.0001; Figure 6E), IL-10 (*p* < 0.0001; Figure 6F), and TNFα (*p* < 0.0001; Figure 6G) in the hippocampus compared to NC-fed mice. We then demonstrate that HFD-fed AKO mice have decreased IFNγ (*p* < 0.0001; Figure 6A), IL-1β (*p* < 0.0001; Figure 6B), IL-5 (*p* < 0.0001; Figure 6C), IL-6 (*p* < 0.014; Figure 6D), KC/GRO(CXCL1) (*p* < 0.0001; Figure 6E), IL-10 (*p* < 0.007; Figure 6F), and TNFα (*p* < 0.0001; Figure 6G) the hippocampus compared to HFD-fed WT mice (Figure 6A–G).

### 2.6. Loss of FABP4 Alleviates Decrease in Microglial UCP2 mRNA Caused by HFD

To determine if loss of FABP4 affects UCP2 mRNA expression under HFD in microglia, we performed a qRT-PCR analysis of UCP2 mRNA from microglia isolated from brain tissues. We demonstrate that HFD-fed WT mice have decreased UCP2 compared to NC-fed mice (*p* < 0.0001; Figure 7). We then demonstrate that HFD-fed AKO mice have increased UCP2 compared to HFD-fed WT mice (*p* < 0.001; Figure 7).

## 3. Discussion

Obesity and neuroinflammation are known risk factors for developing cognitive disorders [26,27,28,29]. HFD induces activation of M1-like microglial phenotypes, leading to impaired immune response and contributing to either the onset of cognitive impairment and/or the acceleration of neurodegenerative diseases such as AD [30]. Herein, we provide evidence that HFD impairs memory, and the microglial FABP4-UCP2 axis is involved in attenuating diet-induced memory impairments.

Microglia are dynamic cells that maintain and promote neuronal health throughout the CNS [31,32]. Neuronal-glial circuitry is maintained via microglia by removing damaged synapses and altering plasticity [33,34]. Depletion of microglia can result in subsequent neuroinflammation and eventual cognitive decline, indicating an important role for microglia in cognition [35]. Previous studies have demonstrated that HFD alters microglial response contributing to cognitive decline [10,36]. Additionally, it has been demonstrated that microglial-mediated synaptic pruning and hippocampal plasticity are impaired via HFD feeding [10]. However, this effect is reversed when the diet is returned to low fat [10], strongly indicating that dietary intake impairs microglial function. We and others have demonstrated that HFD, specifically PA, induces microglial activation [7,20,37]. Our prior data demonstrate that the FABP4-UCP2 axis is central in mediating microglial inflammation [20]. Here, we show that the microglial FABP4-UCP2 axis, in part, mediates protection against diet-induced cognitive decline (Figure 3 and Figure 4).

Despite no differences in learning, long-term spatial memory impairment was evident in WT mice maintained on an HFD as measured by increased latency to find the escape hole (Figure 3). AKO mice maintained on HFD are protected against HFD-induced spatial memory impairment (Figure 3). Importantly, these differences are not attributed to motor deficits, as a similar movement was observed in AKO and WT mice maintained on an HFD (Figure 2). These differences show that HFD-fed mice have impaired spatial memory rather than learning ability. Anxiety-related behavior was tested using an open field system, but no differences were seen between genotypes (see Appendix A). We also demonstrate that short-term working reference memory is impaired in HFD-fed WT mice and that HFD-fed AKO mice are protected against this deficit (Figure 4). These findings are consistent with other reports [38]. It is also important to note that HFD-fed mice have an increase in body fat percentage compared to NC-fed mice for both genotypes. While HFD-fed AKO mice have a slightly lower body fat percentage compared to HFD-fed WT mice, they are still significantly higher than NC-fed AKO mice (Figure 1). Therefore, behavioral differences between groups cannot fully be explained by body fat percentage.

In adipose tissue, FABP4 protein and gene expression are increased during HFD feeding [21,39]. Rodent and clinical data indicate that FABP4 is a significant regulator of SFA-induced inflammation [20,21,39]. The FABP4 knockout model has been shown to confer protection from insulin resistance (IR) despite dietary obesity [23,40]. Additionally, clinical findings support that increased FABP4 serum levels are associated with insulin resistance and secretion in patients with type 2 diabetes mellitus, suggesting that FABP4 plays an important role in peripheral glucose homeostasis (also linked to the onset of AD) [41,42]. Exposure to inflammatory cytokines has also been linked to insulin resistance [43]. Clinically, brain IR is considered an AD risk factor, as patients with AD present increased IR in brain tissue [44]. In HFD rodent models, neuronal IR paired with impaired spatial working memory has been reported [38]. One mechanism in brain tissue could be lipotoxicity, which can impair glucose homeostasis and induce inflammation due in part to the activation of neurotoxic M1 microglial phenotypes, which can precipitate the onset of brain IR [45]. The microglial FABP4-UCP2 axis represents an important mediator in this process, and under HFD conditions, this axis is perturbed, ultimately resulting in cognitive decline. While other groups have determined that FABP5 (E-FABP) knockout mice demonstrate impaired cognitive function, it appears that this is due in part to decreased docosahexaenoic acid (DHA) metabolism within brain tissue [46,47]. A cell-type-specific contribution has not been positively identified, but brain endothelial cells (at the blood and brain barrier), not microglia, are thought to be the most salient contributing cell type [48].

Our prior work demonstrates that loss or inhibition of FABP4 results in reduced microglial pro-inflammatory expression of iNOS and TNF-α via a FABP4-UCP2-mediated axis [20]. Here, we demonstrate that HFD-fed mice have decreased microglial UCP2 mRNA expression compared to NC-fed mice (Figure 7). This decrease is partially alleviated with the knockout of FABP4, as seen in the HFD-fed AKO mice (Figure 7). This is in accordance with our previous work [20]. The decrease in UCP2 expression has been shown in microglia activated to an M1-like pro-inflammatory phenotype, while the increase in UCP2 expression has been shown in microglia activated to an M2-like neuroprotective phenotype [22]. In the absence of UCP2, microglia stimulated with lipopolysaccharide have been shown to produce more nitric oxide and interleukin-6 [22]. We also demonstrate in vivo that HFD-fed WT mice have increased hippocampal inflammatory cytokines and chemokines (IFNγ, IL-1β, IL-5, IL-6, KC/GRO(CXCL1), IL-10, and TNFα; Figure 6A–G), which has been shown by other studies [49,50,51]. This increase is significantly lessened in HFD-fed AKO mice, showing that loss of FABP4 leads to a reduction of hippocampal inflammatory cytokine expression in HFD-fed mice. Of these cytokines and chemokines, IFNγ, IL-1β, IL-6, KC/GRO, and TNFα are known as pro-inflammatory [49,50,51,52]. While IL-10 is known as an anti-inflammatory, it has been shown to increase in the hippocampus after HFD in normal but not diet-resistant animals [50]. These data demonstrate that HFD-fed WT mice have an increase in hippocampal pro-inflammatory cytokine expression, which is lessened with the loss of FABP4. We also demonstrate that HFD-fed WT mice have increased hippocampal microgliosis via increased microglial marker IBA1 expression compared to NC-fed mice (Figure 5). We then demonstrate that HFD-fed AKO mice have decreased hippocampal microgliosis compared to HFD-fed WT mice (Figure 5). Other studies have shown increased IBA1 expression after HFD in the hippocampus [53] and other brain regions [54,55]. The increase in IBA1 is linked to HFD-induced cognitive decline [53,56]. This HFD-induced increase in microgliosis is lessened with the loss of FABP4. This further demonstrates that loss of FABP4 confers protection from neuroinflammation in HFD-fed mice via a mechanism dependent on the microglial FABP4-UCP2 axis.

Multiple studies have demonstrated that lipid metabolism changes are associated with the development of cognitive decline and increased risk of AD [10,12,27,37,57]. In transgenic AD mice (APP23), HFD alters the brain transcriptome, lipid metabolism, and lipid composition, resulting in altered brain immune response [12]. Additionally, HFDs alter phospholipid subspecies linked to neuroinflammation, mitochondrial dysfunction, oxidative stress, and cognitive deficits [12,58]. However, these studies largely focused on whole brain tissue samples and therefore did not reveal individual contributions of microglia. Recent data show that single-cell RNA sequencing (scRNA-seq) is an effective tool for defining microglial immunoheterogeneity in models of AD [59,60]. To identify microglial phenotypes during HFD-induced cognitive decline, single-cell scRNA-seq to perform a comprehensive analysis for identification of genes regulating brain immune system and innate immune response pathways and regulatory factors within the FABP4-UCP2 axis are ongoing. Further work is necessary to understand the role of HFD, microglial lipid metabolism, and cognitive decline. Collectively, our data demonstrate that the microglial FABP4–UCP2 axis alters immune cell metabolism during HFD and obesity. The FABP4-UCP2 axis represents a novel target for the treatment of inflammation-induced neurodegeneration and cognitive decline.

## 4. Materials and Methods

### 4.1. Animals

The FABP4/aP2 knockout (also referred to as AKO) mouse model was originally generated by creating a null mutation in the aP2 gene by homologous recombination, followed by back-crossing onto C57BL6/J mice to obtain homologous null mice. These mice completely lack FABP4 mRNA and protein [23]. Genotyping was performed via qRT-PCR (Transnetyx, Cordova, TN, USA) using primer sequences (see Appendix A). Fifteen-week-old male AKO and wild-type (WT) mice were maintained on 60% high-fat diet (HFD) or normal chow (NC) for 12 weeks [61]. Behavioral testing was performed following 12 weeks of HFD exposure. As our prior studies defined this inflammatory mechanism in male mice, we sought here to confirm whether inflammation affected cognition in the same animals. Future studies are intended to define and examine the same mechanisms in female animals. Mice were obtained from our breeding colony, and the experimental protocol was approved by the Institutional Animal Care and Use Committee at the University of Minnesota.

### 4.2. Body Composition

Body composition was recorded for all groups (n = 22–47) using an EchoMRI 700 (Echo Medical Systems, Houston, TX, USA) before HFD exposure and at 12 weeks on assigned diet [62,63]. Changes in fat mass, lean mass, and body weight were calculated and statistically analyzed.

### 4.3. Locomotor Activity

In order to determine a general assessment of locomotion, animals were removed from the home cage and placed individually into a novel arena (22 × 42 cm; Med-Associates, Fairfax, VT, USA) over a 2 h period (between 1200 and 1600). The chamber is divided into a grid by infrared beams, and an automated system monitors beam breaks to measure exploratory activity [62]. Data are presented as total distance traveled over 2 h.

### 4.4. Barnes Maze

The Barnes maze was used to determine changes in spatial memory as previously described [64]. Mice were placed on a white circular maze (91.44 cm diameter) consisting of 20 evenly spaced holes (7.5 cm apart, 5 cm diameter) located 2 cm from the perimeter. A black escape box, located under one of the holes (defined as target hole), remained in the same place throughout the training period. Distinct spatial cues surrounded the maze and were kept constant throughout the entire experiment. Animals were placed in a dark chamber in the center of the maze for 30 s prior to beginning the task. Once the chamber was lifted, mice were able to use spatial cues to orient themselves and locate the target hole. The session ended when the mouse completely entered the escape box or 3 min elapsed. If the mouse did not enter the hole within the allotted time, it was gently guided into the hole. At the end of each session, the lights were turned off, and the mouse was allowed to stay in the escape box for 10 s before returning to its home cage. Mice were trained for a total of four training sessions per day with at least a 10 min inter-trial interval for a total of four consecutive days. Latency to target hole and average distance from target hole were measured over the 4 training days. On day 5, we tested memory of the escape location (the probe test). The escape hole was removed, and the mouse was allowed to explore the maze for 90 sec. During the probe test, latency to the target hole was recorded [65,66,67,68].

### 4.5. T-Maze Spontaneous Alternation

Spontaneous alternation using the T-maze was used to determine working and spatial reference memory as previously described [38,69]. Briefly, mice were placed in the start box, and the guillotine door was opened after 30 s, and mice were allowed free choice of either arm. Choice is made when the animal’s body completely entered a choice arm, and the animal was contained within the choice arm for 15 s. The mouse was then returned to the start box for 5 s and allowed to again choose either arm for 10 additional trials. Choice arm and alternations were recorded. Percent alternation was calculated as the number of spontaneous alternations (choosing the novel arm, unexplored from the previous trial) divided by the maximum possible alternations (# of alternations/20 × 100%).

### 4.6. Immunohistochemistry

Whole brain was drop-fixed in 10% neutral buffered formalin (Cancer Diagnostics, Durham, NC, USA) for 48 h after tissue collection. This was followed by submersion in 20% sucrose (Sigma, Burlington, MA, USA) in PBS (Thermo Scientific, Waltham, MA, USA) solution for 48 h. Tissue was then stored in cryoprotectant (30% sucrose, 30% ethylene glycol (Sigma, Burlington, MA, USA), 1% polyvinylpyrrolidone 40 (Sigma, Burlington, MA, USA), in PBS) at −20 °C until ready for sectioning. Before sectioning, tissue was submerged in 20% sucrose for 24 h. Sectioning was performed using a sliding microtome, sectioning at 40 μm thickness. Antigen retrieval was performed for 30 min at 78 °C (Reveal Decloaker; Biocare Medical, Pacheco, CA, USA). Blocking of peroxidase was performed for 20 min using 3% H_2_O_2_ (Sigma, Burlington, MA, USA). Sections were blocked for 15 min (100% Background Sniper; Biocare Medical, Pacheco, CA, USA). Sections were incubated overnight in IBA1 primary antibody (1.25 × 10^−5^ mg/mL; Fujifilm Wako, Richmond, VA, USA) solution at 4 °C. Sections were incubated in goat anti-rabbit secondary antibody (2.5 × 10^−4^ mg/mL; Jackson ImmunoResearch, West Grove, PA, USA) for 1 h. Blocking of artificial peroxidase (Vectastain; Vector Laboratories, Burlingame, CA, USA) was performed for 1 h. DAB reaction was performed for 3 min using 4% chromogen (BioLegend, San Diego, CA, USA) in substrate buffer (BioLegend, San Diego, CA, USA). Sections were then mounted and scanned at 40X brightfield (Huron TissueScope LE; Huron Digital Pathology, St. Jacobs, ON, Canada). Densitometry of the hippocampus was performed using Fiji software.

### 4.7. Inflammatory Cytokine Marker Screening

Hippocampal tissue was dissected from whole brain and frozen at −80 °C before homogenization. Tissue was homogenized in RIPA buffer (Thermo Scientific, Waltham, MA, USA) and PIC (HALT; Thermo Scientific, Waltham, MA, USA) using a tissue homogenizer (Bullet Blender; Next Advance, Troy, NY, USA). Protein concentration was determined using an infrared spectrometer (Direct Detect; Millipore, Burlington, MA, USA). A total of 50 μg of protein per sample (n = 7–13) was used for the inflammatory marker assay following manufacturer’s protocol (V-PLEX Proinflammatory Panel 1 (Mouse); IFNγ, IL-1β, IL-2, IL-4, IL-5, IL-6, KC/GRO(CXCL1), IL-10, IL-12p70, and TNFα; MSD, Rockville, MD, USA).

### 4.8. Real-Time PCR

Whole brain tissue was dissected from animals. Microglia were isolated via magnetic-activated cell sorting (MACS) using the Adult Brain Dissociation Kit, mouse, and rat (Miltenyi Biotec, Bergisch Gladbach, Germany) on a gentleMACS Octodissociator with Heaters (Miltenyi Biotec, Bergisch Gladbach, Germany). The cell suspension was labeled with CD11b microbeads (Miltenyi Biotec, Bergisch Gladbach, Germany) and passed through an MS column (Miltenyi Biotec, Bergisch Gladbach, Germany) to select for microglia. Cells were counted using a hemocytometer. Fractions were centrifuged, and the cell pellet was stored at −80 °C until use. Total RNA was extracted from isolated microglia with Trizol (Invitrogen, Waltham, MA, USA) [70,71]. The final concentration of mRNA was determined spectrophotometrically (Nanodrop ND-8000; Thermo Scientific, Waltham, MA, USA). Real-time thermal cycling was carried out in a Roche LightCycler (Roche Diagnostics Corporation, Indianapolis, IN, USA) by one-step qRT-PCR using the general method as previously described [15]. UCP2 gene expression was determined using SYBR Green detection normalized to GAPDH using the Δ−ΔCT method [72]. UCP2 forward primer (5′-3′): TCGGAGATACCAGAGCACTGTCG. UCP2 reverse primer (5′-3′): GCATTTCGGGCAACATTGG. GAPDH forward primer (5′-3′): GACATCAAGAAGGTGGTGAAGCAG. GAPDH reverse primer (5′-3′): AAGGTGGAAGAATGGGAGTTGC.

### 4.9. Statistical Analysis

Statistical differences were determined using one-way ANOVA followed by Tukey’s post hoc test (GraphPad Prism 7; GraphPad, San Diego, CA, USA).

## Figures and Tables

**Figure 1 ijms-23-04354-f001:**
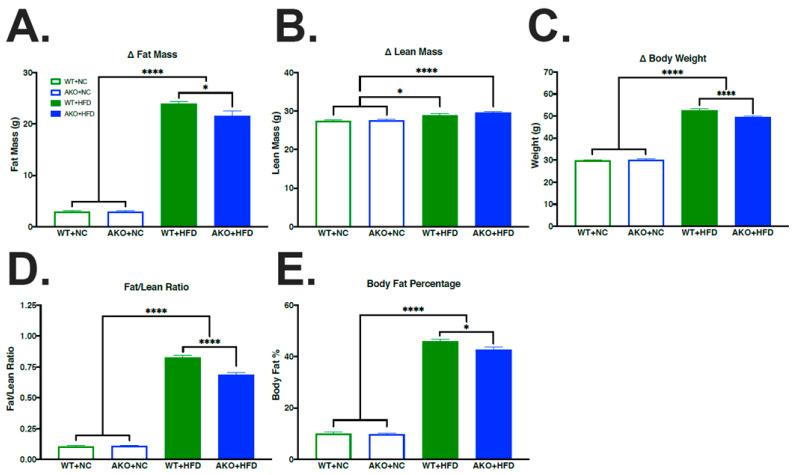
HFD increases fat mass, lean mass, body weight, fat/lean ratio, and body fat percentage. (**A**–**E**) Change in fat mass, lean mass, body weight, fat/lean ratio, and body fat percentage in WT mice fed NC, WT mice fed HFD, AKO mice fed NC, and AKO mice fed HFD (n = 22–47). Data analyzed via one-way ANOVA and Tukey’s post-test. * *p* < 0.05, **** *p* < 0.0001. HFD leads to increase in fat mass (*p* < 0.0001; (**A**)), lean mass (*p* < 0.013; (**B**)), body weight (*p* < 0.0001; (**C**)), fat/lean ratio (*p* < 0.0001; (**D**)), and body fat percentage (*p* < 0.0001; (**E**)). AKO mice fed HFD had decreased fat mass (*p* < 0.044; (**A**)), body weight (*p* < 0.0001; (**C**)), fat/lean ratio (*p* < 0.0001; (**D**)), and body fat percentage (*p* < 0.029; (**E**)) compared to WT mice fed HFD.

**Figure 2 ijms-23-04354-f002:**
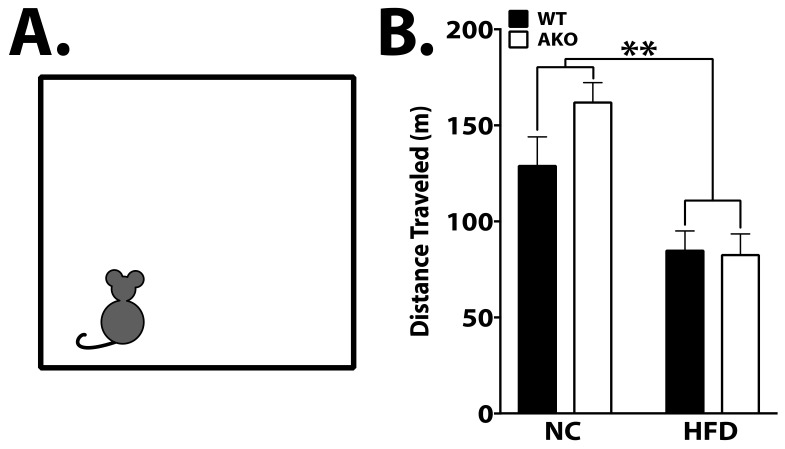
HFD reduces locomotor activity. (**A**) Representation of open field chamber. Mice were placed in a square chamber (27 cm × 27 cm) and allowed to explore freely for 2 h. (**B**) No difference was observed between genotypes, but total distance traveled was significantly reduced in HFD-fed mice (*p* < 0.01). Data analyzed via one-way ANOVA and Tukey’s post-test. ** *p* < 0.01.

**Figure 3 ijms-23-04354-f003:**
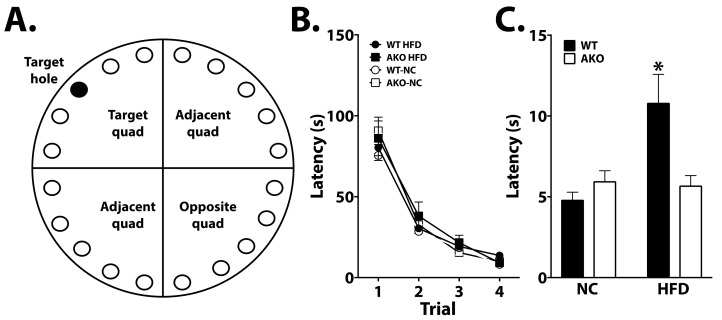
Loss of FABP4 is protective against HFD-induced cognitive decline. (**A**) Representation of Barnes maze apparatus. Training occurred over 4 consecutive days where mice learned to identify the target hole containing an escape box. (**B**) Performance was analyzed for latency to target hole over training days. Performance improved significantly in all groups over duration of training (*p* < 0.0001). (**C**) During the probe (day 5), the escape hole was removed, and latency to target hole was analyzed. HFD-fed WT mice exhibited increased latency to target hole during this test (*p* < 0.05 vs. WT-NC, AKO-NC, AKO-HFD). Data analyzed via one-way ANOVA and Tukey’s post-test. * *p* < 0.05.

**Figure 4 ijms-23-04354-f004:**
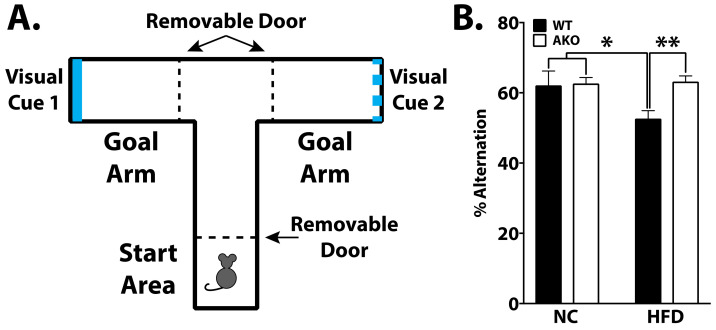
Loss of FABP4 rescues HFD-induced working reference memory decline. (**A**). Representation of T-maze apparatus to measure spatial working reference memory. Mice were placed in start arm and allowed free choice to either goal arm. Once body fully in arm, removable partition was closed allowing mice to stay in goal arm for 15 s and then placed back in start area. Mice underwent an additional 10 trials per day for a total of 2 days. Percent alternation was calculated as the number of spontaneous alternations (choosing the novel arm, unexplored from the previous trial) divided by the maximum number of alternations. (**B**) WT mice fed HFD have reduced alternations over all trials (* *p* < 0.05 vs. WT-NC and AKO-NC, ** *p* < 0.01 vs. AKO-HFD). Data analyzed via one-way ANOVA and Tukey’s post-test.

**Figure 5 ijms-23-04354-f005:**
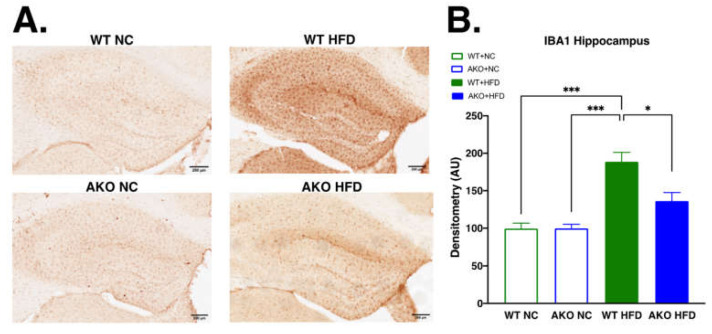
Loss of FABP4 leads to decrease in hippocampal IBA1 expression in mice fed HFD. (**A**) Hippocampal IBA1 stain of WT and AKO mice fed NC or HFD 40X magnification. Scale bar represents 200 μm. (**B**) HFD-fed WT mice had increased hippocampal IBA1 expression compared to NC-fed mice (*p* < 0.001). HFD-fed AKO mice had decreased hippocampal IBA1 expression compared to HFD-fed WT mice (*p* < 0.02). n = 4–5. Data analyzed via one-way ANOVA and Tukey’s post-test. * *p* < 0.05, *** *p* < 0.001.

**Figure 6 ijms-23-04354-f006:**
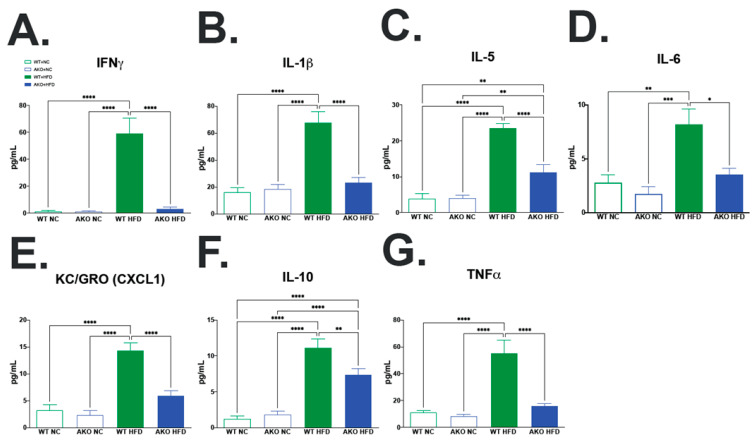
Loss of FABP4 protects against HFD induced increase in hippocampal inflammatory cytokine expression. (**A**–**G**) WT mice fed NC, WT mice fed HFD, AKO mice fed NC, and AKO mice fed HFD (n = 7–13) hippocampal tissue assayed for inflammatory cytokine expression. Expression shown in pg/mL protein, measured via multiplex Meso Scale Discovery assay system. Data analyzed via one-way ANOVA and Tukey’s post-test. * *p* < 0.05, ** *p* < 0.01, *** *p* < 0.001, **** *p* < 0.0001. (**A**) WT Mice fed HFD have increased hippocampal IFNγ expression compared to mice fed NC (*p* < 0.0001), and AKO mice fed HFD (*p* < 0.0001). (**B**) WT Mice fed HFD have increased hippocampal IL-1β expression compared to mice fed NC (*p* < 0.0001), and AKO mice fed HFD (*p* < 0.0001). (**C**) WT Mice fed HFD have increased hippocampal IL-5 expression compared to mice fed NC (*p* < 0.0001), and AKO mice fed HFD (*p* < 0.0001). (**D**) WT Mice fed HFD have increased hippocampal IL-6 expression compared to mice fed NC (*p* < 0.008), and AKO mice fed HFD (*p* < 0.014). (**E**) WT Mice fed HFD have increased hippocampal KC/GRO(CXCL1) expression compared to mice fed NC (*p* < 0.0001), and AKO mice fed HFD (*p* < 0.0001). (**F**) WT Mice fed HFD have increased hippocampal IL-10 expression compared to mice fed NC (*p* < 0.0001), and AKO mice fed HFD (*p* < 0.007). (**G**) WT Mice fed HFD have increased hippocampal TNFα expression compared to mice fed NC (*p* < 0.0001), and AKO mice fed HFD (*p* < 0.0001).

**Figure 7 ijms-23-04354-f007:**
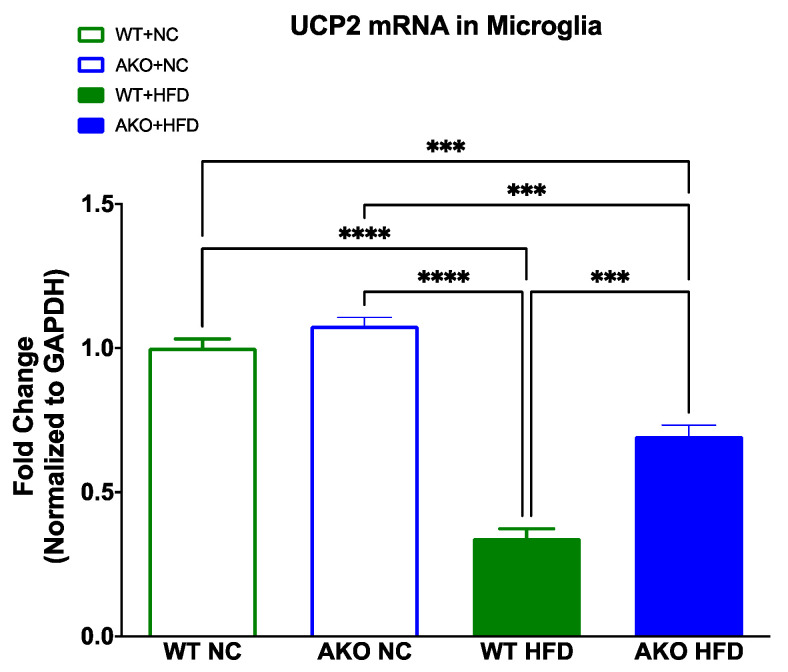
Loss of FABP4 alleviates decrease in UCP2 caused by HFD in microglia. WT mice fed NC, WT mice fed HFD, AKO mice fed NC, and AKO mice fed HFD (n = 3) microglia isolated from brain tissue assayed for UCP2 mRNA expression using qRT-PCR. HFD-fed WT mice had decreased UCP2 compared to NC-fed mice (*p* < 0.0001). HFD-fed AKO mice have increased UCP2 compared to HFD-fed WT mice (*p* < 0.001). Data analyzed via one-way ANOVA and Tukey’s post-test. *** *p* < 0.001, **** *p* < 0.0001.

## Data Availability

Datasets generated during this study are the property of the U.S. Department of Veterans Affairs and will be made available upon request.

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
