# Peer review of "Microglial FABP4-UCP2 Axis Modulates Neuroinflammation and Cognitive Decline in Obese Mice"

_ijms, 2022, doi:10.3390/ijms23084354_

Round 1

Reviewer 1 Report

So et al. have shown that impeding FABP4 function is beneficial in a high-fat diet-induced cognitive decline model. The authors have shown the rescue of cognitive decline in APO KO mice through behavioral assays such as Open field, T-maze, and Barnes maze tests. Immunohistochemistry data has shown a massive reduction in microgliosis, but appropriate controls are missing. Relevant statistical analysis is missing or unlabelled, hindering inference on these results. The manuscript is well written and has some potentially clinically relevant results. However, the authors have failed to provide ample evidence to support their conclusion.    Major comments   1. It is known that a high-fat diet induces cognitive decline leading to dementia and even AD. Insulin resistance has been one of the major causes of cognitive decline in a high-fat diet. Although authors have mentioned the possibility of Insulin resistance as pathological in their model, they have not provided any evidence. Meanwhile, the authors have not provided any other evidence regarding how FABP4 knockout mice demonstrate rescue against cognitive dysfunction after a high-fat diet. FABP4-UCP2 axis is attributed to regulating cognitive decline and is mentioned throughout the title and the manuscript's text. However, data on UCP2 is missing.   2. P values are missing (figure 1, 6) in several figures and text. Statistics model missing in figure legend (figure 2).   3. The authors hypothesize that FABP4 attenuates HFD-induced cognitive decline via reduced neuroinflammation. However, the authors have not sufficiently characterized HFD induced neuroinflammation in this manuscript.   4. Why does AKO (NC) mouse travel more distance than WT (NC). Could this be a sign of anxiety behavior?   5. There is inadequate information regarding the FABP4 knockout mice model in the method/materials section.   6. Figure 5. The absence of control for WT HFD and AKO HFD is not optimal.    Minor comments   - The quality of the figures should be improved and be consistent. Figure label size is not consistent   - Figure 4. It will benefit the reader if the term "T-maze alteration" is defined in the result section.   - Figure 2. Representation of a field chamber is unnecessary and could be excluded.

Reviewer 2 Report

In this report, the authors hypothesized that the FABP4-UCP2 axis is involved in regulating diet-induced cognitive decline. This team analyzed cognitive function, hippocampal microgliosis, and cytokine expression in mice lacking microglial FABP4 15 using AKO mice. Their results support that loss of FABP4 prevents cognitive decline in vivo. This is a series of research the authors have been performing. This issue is novel and valuable, and the manuscript was well written. There are some suggestions listed as follows.

  1. Firstly, the protein and gene level of FABP4 should be verified in AKO mice.
  2. HFD-fed mice seem not to increase lean mass compared to NC-fed mice, based on figure 1B. “2.1 HFD increases fat mass, lean mass,……”? The symbol of the p-value was not labeled in the bar chart.
  3. In the results of cognitive decline, the learning ability had no impairment, but long-term spatial memory impairment was noticed in WT mice maintained on a HFD. These results may be not reasonable. Please make an explanation and discussion for these findings. And the data of time spent in the quadrant containing the target hole and the average distance from the target hole are not available.
  4. The t-test was neglected in the section “Statistical Analysis.”
  5. The IBA1 expression in NC-fed mice (baseline) should be displayed in figure 5.
  6. The statement “HFD-fed AKO mice have a slightly lower body fat percentage compared to HFD-fed WT mice” cannot be consistent with Figure 1A. There is no significant difference based on the present bar chart.
  7. The title mentioned about microglial FABP4-UCP2 axis, but the role of UCP2 or ROS in this in vivo model was not verified.

Round 2

Reviewer 2 Report

To easily read, the label "*" could be inserted into the bar chart in Fig.1&6.